# *miR-9a* modulates maintenance and ageing of *Drosophila* germline stem cells by limiting N-cadherin expression

Yehonatan Epstein[1], Noam Perry[1], Marina Volin[1], Maayan Zohar-Fux[1], Rachel Braun[1], Lilach Porat-Kuperstein[1] & Hila Toledano [1]

Ageing is characterized by a decline in stem cell functionality leading to dampened tissue regeneration. While the expression of microRNAs across multiple species is markedly altered with age, the mechanism by which they govern stem cell-sustained tissue regeneration is unknown. We report that in the *Drosophila* testis, the conserved *miR-9a* is expressed in germline stem cells and its levels are significantly elevated during ageing. Transcriptome and functional analyses show that *miR-9a* directly regulates the expression of the adhesion molecule N-cadherin (N-cad). *miR-9a* null mutants maintain a higher number of stem cells even in the aged tissue. Remarkably, this rise fails to improve tissue regeneration and results in reduced male fertility. Similarly, overexpression of N-cad also results in elevated stem cell number and decreased regeneration. We propose that *miR-9a* downregulates *N-cad* to enable adequate detachment of stem cells toward differentiation, thus providing the necessary directionality toward terminal differentiation and spermatogenesis.

[1] Department of Human Biology, Faculty of Natural Sciences, University of Haifa, 199 Aba Hushi Avenue, Mount Carmel, Haifa 3498838, Israel. Correspondence and requests for materials should be addressed to H.T. (email: hila@sci.haifa.ac.il)

Ageing leads to reduced tissue homeostasis and a decline in the ability to replace damaged cells by new functional ones[1]. Homeostasis and repair of many adult tissues, such as blood, gut, and testis, are supported by small specialized populations of tissue-specific stem cells. In a given tissue, stem cells reside in a local microenvironment (niche) that acts as a control-unit to determine stem cell proliferation rate and protects the overall stem cell pool from depletion[2].

In the *Drosophila melanogaster* (*Drosophila*) testis, mature sperm cells are generated by germline stem cells (GSCs) that are located at the apical tip of the testis. These cells, together with Cyst stem cells (CySCs) co-habitat in the niche and adhere around a cluster of somatic cells called the hub[3–5]. The hub is a spherical three-dimensional (3D) structure of approximately 12 cells, the great majority of which (Fig. 1a) (~9–10) are associated on several planes with all the surrounding stem cells. The GSCs are in direct contact with the hub via microtubule-based

nanotubes that protrude directly into the hub[6]. The hub expresses signaling and adhesion molecules that maintain the stem cells within the niche. Upon GSC division, one of the two daughter cells remains adherent to the hub for self-renewal, while the other, a displaced progenitor cell, undergoes transit amplification divisions to generate spermatogonia progenitor cells before becoming a terminally differentiated spermatocyte[7]. GSCs have an age-dependent limited half-life of 14 days and are lost mainly via detachment from the niche[8]. In contrast, spermatogonia cells can dedifferentiate back into GSCs during ageing to repopulate the niche (Fig. 1a)[9, 10], thus representing an intermediate cell population that can either differentiate or dedifferentiate according with the needs of the tissue. Therefore, although ageing results in significantly smaller testis with reduced cells of all types, unlike females, aged males (30-days) tend to remain fertile[7, 11, 12]. Nonetheless, since all testicular germ cells originate from a minute number of stem cells (approximately seven) even a small

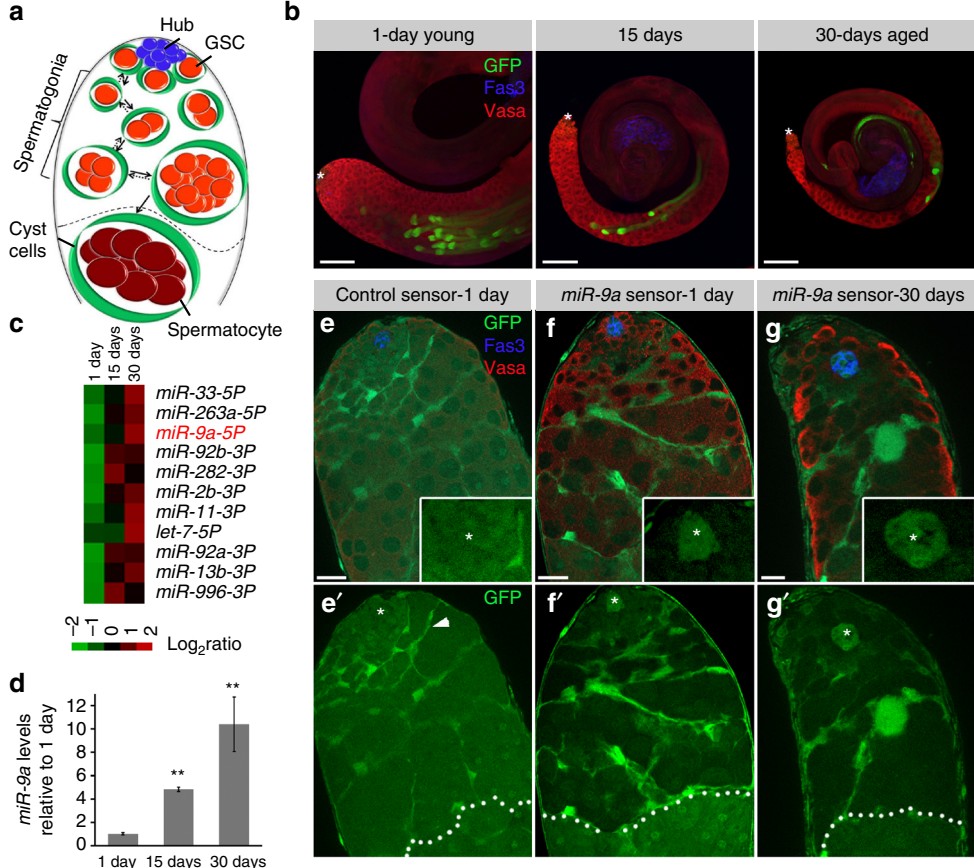

**Fig. 1** *miR-9a* increases during ageing and is expressed in GSCs and spermatogonia. **a** Side-view representation of the apical tip of the *Drosophila* testis. GSCs (*red*) are anchored to the hub (*blue*) and spermatogonia germ cells (*red*) can terminally differentiate to spermatocytes (*crimson*) or dedifferentiate to GSCs (*two-directional arrows*). GSCs and spermatogonia are encapsulated by cyst cells (*green*). *Dashed-line* separates spermatogonia from spermatocytes. **b** Testes from 1-day, 15-days, or 30-days-old males (*green*, mature sperms; *Imp^CB04573*) immunostained for Vasa (*red*, germ cells) and Fas3 (*blue*, hub and seminal vesicles). Note an age-related decrease in cell numbers and in overall testis size. *Scale bars*, 100 μm. *Asterisks* mark the hub. **c** Heatmap of fold-change of *Drosophila* miRNAs in testes of 1-day, 15-days, or 30-days-old wild-type (*w^1118^*) flies. Top 11 most age-altered miRNAs out of a total of 100 are shown. DESeq defined significance between 30-days and 1-day for each miRNA ($P_{adj} < 0.05$; $n = 2$ replicas for each time point). **d** qRT-PCR for mature *miR-9a* relative to control (*2S rRNA*) in the testes of 1-day, 15-days, or 30-days-old wild-type (*w^1118^*) males. Levels are normalized to 1-day-old adults. *Error bars* denote s.d. of three biological repeats each in triplicate measurements. Note a 10-fold increase of *miR-9a* in 30-days. Statistical significance was determined by one-way ANOVA and post hoc analysis was performed with Tukey multi-comparison test. **$P \le 0.005$ between 15 and 1-day and between 30 and 1-day. **e–g'** Testes of control GFP sensor (**e**, **e'**) and *miR-9a* sensor 1-day (**f**, **f'**) and 30-days (**g**, **g'**) stained for Fas3 (*blue*) to mark the hub (*asterisks*), Vasa to mark the germ cells (*red*) and GFP (*green*). A control sensor is expressed in all cells at the apical tip of the testis including: hub, GSCs, CySCs, cyst cells (*arrowhead*), spermatogonia and spermatocytes. The *miR-9a* sensor detect endogenous levels of *miR-9a* in GSCs and spermatogonia. *Dashed-line* separates spermatogonia from spermatocytes. Note that *miR-9a* is not expressed in spermatocytes or the somatic niche (hub and cyst cells). All images in all figures are single sections. *Scale bars*, 10 μm

change in their overall number or division frequency is expected to impact tissue regeneration.

MicroRNAs (miRNAs) are negative regulators of mRNA targets that prevent their translation into proteins. Recent data reveals that the levels of some miRNAs change with age across organisms, yet little is known about the molecular pathways that are regulated by these changes[13]. Here, we aimed to identify the miRNAs that change in the course of ageing and to find how these changes affect stem cell functionality. We found that *miR-9a* levels increase in the GSCs during ageing. Furthermore, *miR-9a* directly downregulates Neural-Cadherin (N-cad) to control the adhesion between GSCs and the hub, thus promoting GSCs detachment from the niche to allow differentiation and functional spermatogenesis.

## Results

**miR-9a levels increase in testis of aged males**. To determine whether the levels of specific miRNAs are age-altered, we analyzed the miRNAome of testes dissected from 1-day (young), 15-days (mid-aged), and 30-days (aged) wild-type (*w1118*) flies by NanoString technology (Fig. 1b, c). The resulting miRNAome showed that of a total of approximately 100 miRNAs, the expression level of a small cohort of 11 was elevated by more than two-folds in aged flies (Fig. 1c). The cohort included *let-7* that was found to regulate GSCs niche ageing, thus supporting library reliability[12]. Interestingly, one of the top candidates identified was the evolutionary conserved *miR-9a*[14–17]. *miR-9a* was also included in the top four most abundant miRNAs in the testis and in aged flies represents ~1% of the entire miRNAome. A subsequent qRT-PCR analysis revealed that compared to young males, *miR-*

*9a* levels were increased by five-fold in testis of mid-aged males and by ten-fold in aged males (Fig. 1d).

**miR-9a is expressed in GSCs and progenitor germ cells**. To identify the cells that express *miR-9a* in the testis we used a green fluorescent protein (GFP) sensor that utilizes the unique property of miRNAs to silence protein expression[18]. The GFP-*miR-9a* sensor contains two repeats of the complementary sequence of *miR-9a* in an artificial 3′ untranslated region (3′UTR) following a reporter GFP sequence[15]. Therefore, cells that endogenously express *miR-9a* create a silencing mechanism that prevents the expression of GFP. We used this method to compare the expression pattern of GFP-control and GFP-*miR-9a* sensor both driven under a tubulin promoter. GFP of the control sensor is expressed in all the cells of the testis, with a brighter signal in cyst cells as expected from high Tubulin expression in these long thin cells (Fig. 1e). However, in testes of both young and aged flies, the *miR-9a* sensor revealed that *miR-9a* is expressed in GSCs and spermatogonia progenitor germ cells. Moreover, *miR-9a* is not expressed in terminally differentiated spermatocytes, mature sperms, somatic cyst, and, notably, not in hub cells (Fig. 1f, g). Expressing *miR-9a* sensor in a *miR-9a* null mutant background of *miR-9a[E39]*[14] resulted in GFP expression in GSCs and spermatogonia cells, indicating that in the sensor flies GFP is specifically repressed by *miR-9a* in these cells (Supplementary Fig. 1a, b). In support of these data, *miR-9a* fluorescence in situ hybridization (FISH) shows that *miR-9a* is expressed in GSCs and spermatogonia and absent from the hub. Moreover, no *miR-9a* FISH signal was obtained in testis of *miR-9a[E39]* null mutants (Supplementary Fig. 1c–e).

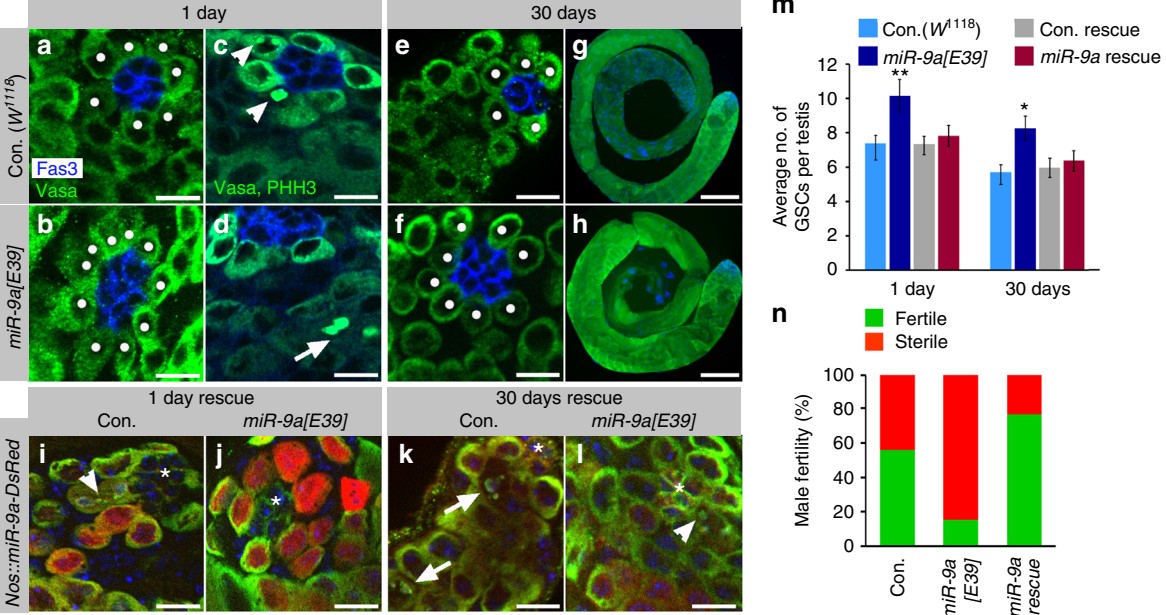

**Fig. 2** *miR-9a* mutants increase GSCs maintenance but reduce spermatogenesis. **a–h** Testes from 1-day (**a**, **c**) or 30-days (**e**, **g**)-old control (*w1118*) or 1-day (**b**, **d**) or 30-days (**f**, **h**)-old *miR-9a[E39]* mutants immunostained for Fas3 (*blue*, hub), Vasa (*green*, germ cells), and with pHH3 (*green*, **c–d**). *White dots* denote GSCs. **i–l** Rescue of *miR-9a[E39]* mutants by ectopic expression of *DsRed-miR-9a* in GSCs and progenitor germ cells (*nos-GAL4, UAS-DsRed-miR-9a*; *miR-9a[E39]*), Con. rescue (*nos-GAL4, UAS-DsRed-miR-9a*; *miR-9a[E39]*/TM6), young (**i**) and aged (**k**); Vasa, Fas3, and pHH3 (*green*) DsRed (*red*) and DAPI (*blue*). *Asterisks* mark the hub, *arrowhead* denote mitotic GSCs and *arrow* mitotic spermatogonia. *Scale bars* 10 μm (**a–f**, **i–l**) and 100 μm (**g–h**). **m** Shown are average number of GSCs per testis along with 95% confidence interval (*error bars*). The total number of testes scored: con. (*w1118*) 1-day (*n* = 47), 30-days (*n* = 60); *miR-9a[E39]* 1-day (*n* = 71), 30-days (*n* = 70); Con. rescue 1-day (*n* = 45), 30-days (*n* = 24); *miR-9a[E39]* rescue 1-day (*n* = 33), 30-days (*n* = 16); Statistical significance was determined as in Fig. 1d. **P ≤ 0.005, *P ≤ 0.01 between *miR-9a[E39]* and the other three genotypes (Con. *w1118*, Con. rescue, and *miR-9a [E39]* rescue) of the same age. **n** Fertility assay of aged males (30 days) from control (*w1118*, *n* = 36) and *miR-9a[E39]* mutants (*n* = 41), *miR-9a[E39]* rescue (*n* = 20). Note decreased fertility of aged *miR-9a[E39]* mutants and rescue in *DsRed-miR-9a* ectopic expression in GSCs and spermatogonia of the mutant background

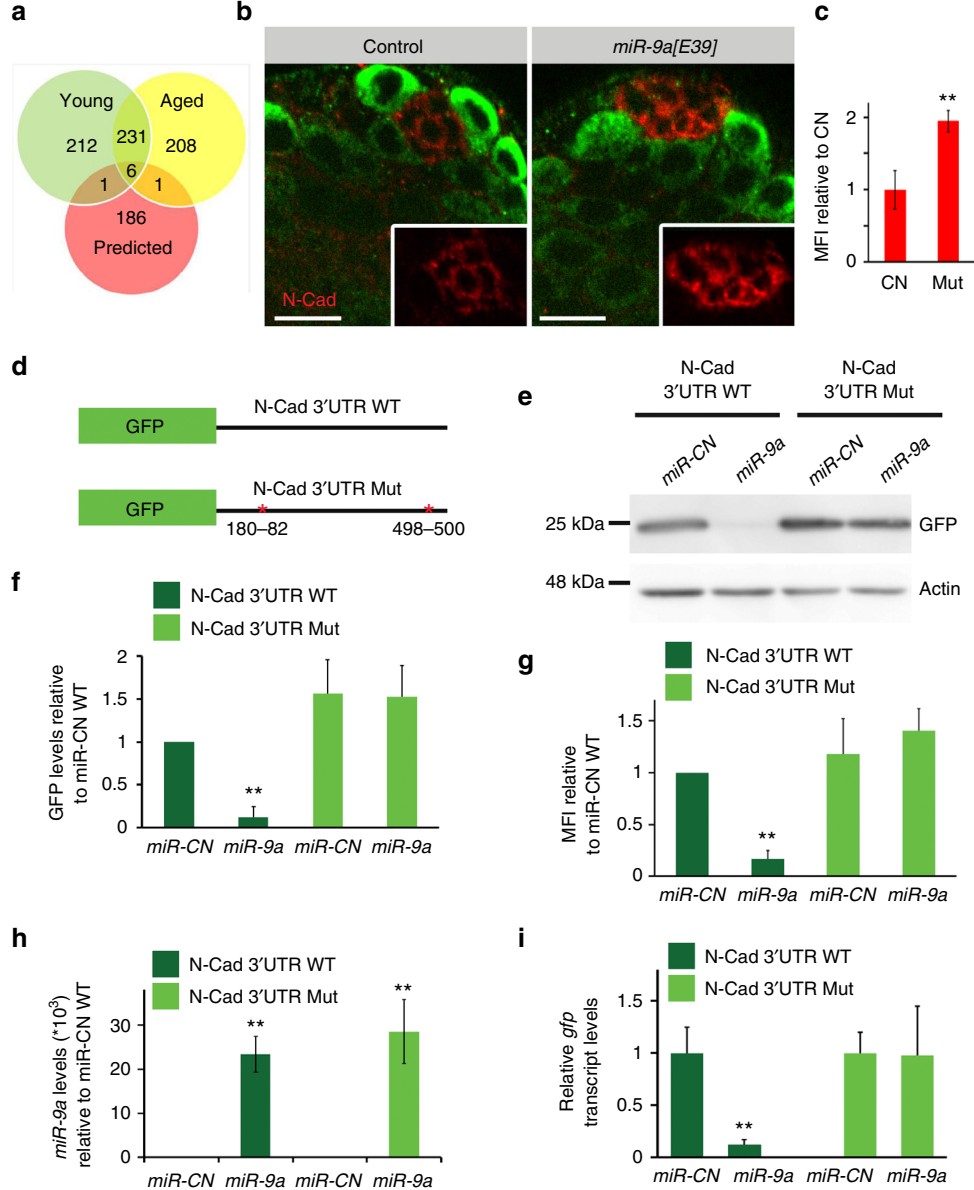

**Fig. 3** *N-cad* is a direct target of *miR-9a*. **a** Venn diagram of genes increased in testis from young (*green*) or aged (*yellow*) *miR-9a[E39]* mutants relative to age-matched controls transcriptome analysis in compare to computationally predicted *miR-9a* targets (*red*; http://www.targetscan.org/). **b** Representative images of testes taken at the same time exposure from control (*w[1118]*; n = 45) and *miR-9a[E39]* mutants (n = 45) immunostained for N-cad (*red*) and Vasa (*green; scale bars*, 10 μm). **c** Immunofluorescence signal quantification. Note the significant increase in N-cad expression in GSCs-hub adherent junction of *miR-9a[E39]* mutants. *P*-values were generated after a two-tailed Student's *t*-test (**P < 0.005). **d–i** *miR-9a* targets a *gfp-N-cad*-3′UTR reporter in S2R+ cells. **d** Schematic representation of GFP reporter constructs of N-cad 3′UTR WT or Mut (mutated bases to disrupt *miR-9a* seed pairing are indicated). **e–f** Western blot analysis of cells transfected with 50 nM *miR-9a* mimic or negative control miRNA (*miR-CN*) with GFP reporters of N-cad 3′UTR WT or Mut as indicated. **e** Shown is a representative immunoblot of n = 3 (biological repeats). **f** GFP quantification relative to actin (n = 3, biological repeats). **g** MFI of GFP from the GFP-positive cells was analyzed by flow cytometry (n = 3, biological repeats). **h,i** qRT-PCR, average of n = 3, biological repeats (each in three replicas) for (**h**) mature *miR-9a* levels relative to *miR-CN* in N-cad 3′UTR WT. *P*-values were generated after a two-tailed Student's *t*-test (**P < 0.005). **i** *gfp* transcript levels relative to *miR-CN* in N-cad 3′UTR WT

***miR-9a* null present higher GSCs number with reduced division**. To define the function of *miR-9a* in stem and progenitor germ cells, we compared the niche of young and aged *miR-9a [E39]* mutant males to age-matched controls. Immunofluorescence microscopy with germ and hub cell markers was used to evaluate the number of GSCs that are defined as germ cells that are physically attached to the hub[11]. Our analysis showed that compared to control (*w[1118]*), *miR-9a[E39]* young mutants contain 37% more GSCs in their niche (Fig. 2a, b, m). Furthermore, these differences were maintained also in aged

males, where *miR-9a[E39]* mutants showed 45% more GSCs (Fig. 2e, f, m). Similar to *miR-9a[E39]*, a second *miR-9a* null allele, *miR-9a[j22]*[14], also maintains a high average number of GSCs in the niche of young and aged males (Supplementary Fig. 1f).

To our surprise, the marked increase in GSCs number in *miR-9a[E39]* mutants did not result in improvement of the ageing phenotype. Similar to age-matched controls, testes of aged *miR-9a[E39]* mutants remained small, and presented with an even worse fertility phenotype; 85% *miR-9a[E39]* were sterile

**Table 1 Six potential mRNA targets for *miR-9a* in the testis**

| Gene | 1-day young | | | | | 30-days aged | | | | | Seeds |
|---|---|---|---|---|---|---|---|---|---|---|---|
| | logFC | Significance (*P*-value) | Average CPM per gene | | | logFC | Significance (*P*-value) | Average CPM per gene | | | |
| | | | WT | *Mir-9a* mutant | | | | WT | *Mir-9a* mutant | | |
| ***N-cad herin*** | 1.6 | $1.3E^{-26}$ | 20.7 | 63.4 | | 0.9 | $4.9E^{-08}$ | 27.0 | 49.3 | | 2 |
| *sticks and stones* | 1.2 | $1.08E^{-05}$ | 3.08 | 7.1 | | 1.0 | 0.0005 | 2.57 | 5.23 | | 1 |
| *CG10512* | 1.5 | $5.25E^{-25}$ | 23.56 | 67.0 | | 1.5 | $1.5E^{-22}$ | 41.29 | 115.62 | | 1 |
| *lame duck* | 2.0 | $1.26E^{-09}$ | 1.02 | 4.2 | | 1.1 | 0.0009 | 1.6 | 3.43 | | 1 |
| *meso18e* | 1.3 | $7.35E^{-11}$ | 9.79 | 23.6 | | 1.0 | $1.6E^{-6}$ | 11.9 | 23.5 | | 1 |
| *CG34136* | 1.5 | 0.0002 | 1.03 | 3.0 | | 2.2 | $1.6E^{-6}$ | 0.7 | 3.1 | | 1 |
| *Senseless* | 1.0 | 0.0001 | 3.8 | 7.7 | | 1.0 | 0.001 | 3.5 | 6.4 | | 1 |
| *E-cad herin* | 0.1 | 0.453 | 54.5 | 59.3 | | 0 | 0.95 | 63.4 | 64.1 | | 0 |

Transcriptome filtration based on logFC ≥ 0.9, significance cutoff (*P*-value ≤ 0.05), and minimal reading levels (sum reads per each gene ≥ 1) obtained a group of 231 genes that showed higher expression in *mir-9a[E39]* mutant vs. control in either young or aged testis. Comparison of this list to the predicted *miR-9a* targets resulted in the presented six candidates. The negative control, *E-cad*, is shown at the bottom of the table.

compared to 40% control (Fig. 2g, h, n). To determine why the increase in GSCs of *miR-9a[E39]* mutants does not improve spermatogenesis, we immunostained testes with anti-Thr 3-phosphorylated histoneH3 (pHH3) to mark mitotic cells (Fig. 2c, d) and counted pHH3-positive GSCs. In agreement with previous findings, the division frequency of young control GSCs was decreased from ~6% (13/223 GSCs) to ~3% (7/258 GSCs) in aged flies[9]. However, GSCs of *miR-9a[E39]* and *miR-9a[J22]* mutant alleles completely arrested division in aged flies ($n = 233$ and $n = 178$, respectively). Thus, although the niche of *miR-9a* null mutants consists of a higher number of GSCs, these cells fail to maintain spermatogenesis in aged males due to reduced division frequency.

To determine whether these mutant phenotypes are due to lack of *miR-9a* in GSCs and spermatogonia cells, we ectopically expressed *UAS-DsRed-miR-9a*[15] in these cells of *miR-9a[E39]* mutants (*nos-GAL4, UAS-miR-9a-DsRed; miR-9a[E39]*). DsRed fluorescent signal was used to mark the *miR-9a*-positive cells (Fig. 2i–l). This ectopic expression was sufficient to return the average number of GSCs associated with the hub back to normal numbers both in young and aged adults (Fig. 2m). Moreover, overexpression of *miR-9a* in *miR-9a[E39]* mutant regained GSCs division frequency to 3% ($n = 102$) and rescued fertility of the aged mutant males (Fig. 2n).

**Identification of *miR-9a* targets in the testis**. miRNAs repress mRNA translation, which is often followed by the mRNA deadenylation and decay[19]. Thus, the mRNA levels of direct *miR-9a* targets in the testis are expected to be elevated in *miR-9a[E39]* mutants. To facilitate *miR-9a* target identification we analyzed the transcriptome of cDNA libraries (Illumina) of four RNA samples (each in at least two biological repeats) prepared from testis of young (1-day), aged (30-days), control (*w1118*), and *miR-9a[E39]* mutants. Reads were aligned to the *Drosophila* genome and gene expression levels were quantified using Htseq-count. This provided a list of 11,416 genes that are expressed in the testis (Supplementary Fig. 2). Differential gene analysis using edgeR-classic method provided count per million (CPM) values and *P*-values. After filtration based on log Fold Change (logFC ≥ 0.9), significance cutoff (*P*-value ≤ 0.05), and minimal CPM per each gene (≥1), we obtained a group of 450 genes that showed higher expression in young *mir-9a[E39]* mutant vs. control, and 446 genes that increased in old mutants vs. control. Of these, 231 genes showed higher expression in *miR-9a[E39]* mutant vs. control in both young and aged testis (Fig. 3a). A comparison of this list to the 194 in silico predicted *miR-9a* targets (Targetscan Fly) yielded six potential direct targets, one of which was *senseless*,

a previously characterized target of *miR-9a*, confirming library reliability (Fig. 3a and Table 1)[14, 17]. However, antibody staining did not reveal the presence of senseless in the testis. Notably, the list also included *N-cad*, a typical member of the cadherin family of proteins that forms $Ca^{+2}$-dependent homophilic interactions to mediate dynamic cell–cell adherent junction[20]. N-cad levels were significantly higher in testis of both young and aged *miR-9a[E39]* mutants compared to age-matched controls (Table 1). In contrast, the levels of *Epithelial-cadherin* (E-cad), an N-cad family member that is not a predicted *miR-9* target, were unchanged (Table 1). At the apical tip of the testis N-cad is expressed both in the adherent junctions that connect hub cells to each other and in the junctions that connect GSCs to the hub[11, 21].

**Validation of *N-cad* as a *miR-9a* target**. Consistent with the transcriptome analysis, immunofluorescence staining of controls and *mir-9a[E39]* mutants with anti-N-cad revealed higher levels in adherent junction of mutants, among hub cells and between hub and GCSs (Fig. 3b). Examination of image Z-sections and a 3D projection revealed that the majority of hub cells generate N-cad boundaries with GSCs in several planes (Supplementary Fig. 3a, b). Therefore, what may appear as an increase in N-cad among hub cells following *miR-9a* knockdown is in fact an increase that occurs mainly between hub and GSCs. Signal quantification of images taken at the exact same exposure showed a 1.9-fold increase of N-cad in *mir-9a[E39]* mutants compared to controls (Fig. 3c). In accordance with this observation, reducing *N-cad* levels in GSCs of the *mir-9a[E39]* mutant flies resulted in an overall reduction of N-cad expression (Supplementary Fig. 3c, d).

Two *miR-9a* canonical recognition sites are located within 3′ UTR of *N-cad* (Targetscan Fly, Table 1 and Fig. 3d). Western analysis in Schneider 2 (S2R+) cells showed that *miR-9a* causes a 90% reduction in the expression of a GFP reporter containing the *N-cad* 3′UTR (*gfp-N-cad-3′UTR^WT*). This effect was abolished by mutating the predicted target sites (*gfp-N-cad-3′UTR^Mut*), indicating that seed mutations rendered the reporter resistant to translation inhibition (Fig. 3d–f). Consistent with these observations, flow cytometry (FACS) analysis of the GFP-expressing cells population revealed a marked 83% decrease in the mean fluorescent intensity (MFI) when *miR-9a* was co-expressed with *gfp-N-cad-3′UTR^WT* reporter (Fig. 3g). The transfection efficiency of *miR-9a* co-expression with *gfp-N-cad-3′UTR^WT* or *gfp-N-cad-3′UTR^Mut* reporters was measured by qRT-PCR for mature *miR-9a*, and confirmed similar expression levels (Fig. 3h). Moreover, *miR-9a* overexpression did not affect cell viability. Quantification of *gfp* by qRT-PCR revealed a 90% reduction in mRNA levels of

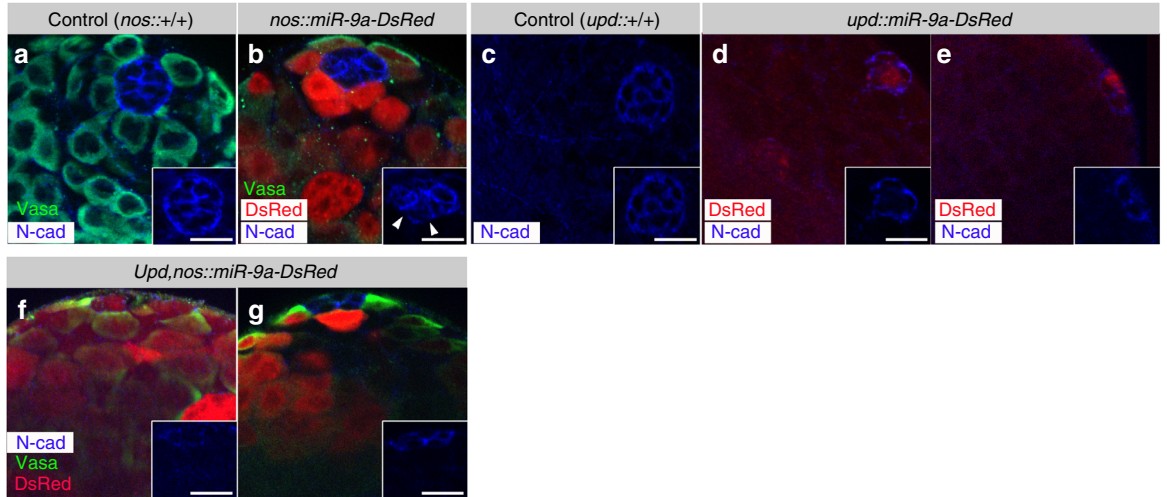

**Fig. 4** *miR-9a* overexpression reduces N-cad. Ectopic expression of *DsRed-miR-9a* in GSCs and progenitor germ cells (**a**, **b**) or the hub (**c–e**) or in both the hub and GSCs (**f**, **g**) reduces N-cad expression. Testes detected for DsRed (*red*) and were stained with N-cad (*blue*) and Vasa (*green*). **a**, **b** Control (*nos-GAL4* outcrossed to *w[1118]*) and *miR-9a* overexpression in the GSCs and progenitor germ cells (*nos-GAL4,UAS-DsRed-miR-9a*). Note N-cad reduction between GSCs and hub cells (**b** and inset). **c–e** Control (*upd-GAL4* outcrossed to *w[1118]*) and *miR-9a* overexpression in the hub (*upd-GAL4;UAS-DsRed-miR-9a*). Note N-cad reduction between hub cells (**d**, **e**, and insets). **f**, **g** Simultaneous expression of *miR-9a* in both the hub and GSCs (*upd-GAL4;nos-GAL4,UAS-DsRed-miR-9a*). 36% of the testes showed *DsRed-miR-9a* both in the hub and germline (**f**, *n* = 39) and 64% showed *DsRed-miR-9a* only in the germline (**g**). Note a dramatic reduction of N-cad among hub cells and between hub and GSCs (**f** and inset). *Scale bars* 10 μm

the wild-type reporter only when *miR-9a* was co-expressed, with no effect on the mutant (Fig. 3i). These data confirm that *miR-9a* directly inhibits N-cad protein expression and destabilizes its mRNA through canonical sequences within the 3′UTR.

**Misregulation of *miR-9a* in the stem cell niche**. Given that *miR-9a* is expressed in GSCs and not in the hub, we tested the possibility that it regulates the adherent junctions between GSCs and hub cells via N-cad levels. We used *UAS-miR-9a-DsRed*[15] to overexpress *miR-9a* either in GSCs and spermatogonia cells (*nos-GAL4; UAS-miR-9a-DsRed*) or in the hub (*upd-GAL4; UAS-miR-9a-DsRed*), whereas the *DsRed* fluorescent signal marks the positive-expressing cells. *miR-9a* overexpression in GSCs and spermatogonia cells that normally express *miR-9a* resulted in fragmented N-cad expression in the adherent junctions between GSCs and hub cells (Fig. 4a, b). Moreover, ectopic expression of *miR-9a-DsRed* in hub cells, which do not express *miR-9a* in wild-type, was partial as only some of the testes expressed DsRed. However, all the samples where DsRed was detected (8/21) showed a dramatic decrease in N-cad expression in adherent junctions among hub cells (Fig. 4c–e). Simultaneous expression of *UAS-miR-9a-DsRed* in both the hub and GSCs (*upd-GAL4; nos-GAL4, UAS-miR-9a-DsRed*) also caused a dramatic reduction in N-Cad expression (Fig. 4f). However, here too, clear detection of the DsRed signal in both hub and GSCs was only apparent in a few samples (14/39), while the rest of the samples expressed *miR-9a-DsRed* only in the germline (25/39, Fig. 4g). Together, these data suggest that *miR-9a* expression in GSCs regulates the dynamic adhesion between these cells and the hub.

**N-cad overexpression in germline stem and progenitor cells**. To determine whether N-cad overexpression presents a similar phenotype to that of *miR-9a* mutants, we measured the stem cell number, division frequency, and fertility of N-cad overexpressing flies in stem and spermatogonia germ cells (*UAS-N-cad; nos-GAL4*) compared to controls (*nos-GAL4* outcrossed to *w[1118]*). N-cad overexpression in young and aged males resulted in a significantly higher number of stem cells that were attached to the niche (Fig. 5a–e). However, GSC division frequency as measured

by pHH3-positive cells was markedly reduced (0.7%) in aged N-cad overexpressing males (*n* = 277). Fertility tests showed that 45% of these flies are sterile compared to 19% of controls (Fig. 5f). Taken together, these findings indicate that the phenotype of *miR-9a* mutants origins at least partially from N-cad over-expression in stem cells.

***N-cad[RNAi]* in GSCs rescues the *miR-9a[E39]* phenotype**. Lastly, to directly address the possibility that the *miR-9a[E39]* mutant phenotypes of increased GSCs and decreased fertility are due to the presence of elevated N-cad at the hub-GSC junctions, we reduced its expression in GSCs of the mutants (*nos-GAL4, UAS-N-cad[RNAi]; miR-9a[E39]*, Fig. 5g–l). This restored normal GSC number in both young and aged adults (Fig. 5k), regained normal division rate of GSCs in aged males (*n* = 178), and importantly rescued the age-related sterility of *miR-9a[E39]* mutants (Fig. 5l).

**Discussion**

In summary, our results suggest that *miR-9a* levels increase significantly in the testis during ageing to promote stem cell differentiation and detachment from their niche. While both young and aged *miR-9a* mutants hold a much higher number of stem cells in their niche, this does not improve spermatogenesis. On the contrary, GSC division frequency is reduced and spermatogenesis of *miR-9a* mutants is gradually decreased leading to premature sterility in aged males.

The appearance of high levels of N-cad in the niche of the *miR-9a* null mutants may be explained by the fact that GSCs are directly connected to the majority of the hub cells. Alternatively, it is possible that N-cad expression in the hub is non-autonomously regulated by N-cad levels or by other *miR-9a* targets in the GSCs. We present several lines of evidence to support that *N-cad* is a direct target of *miR-9a*. Two *miR-9a* binding consensus sites are located at the 3′UTR of *N-cad* to mediate translation inhibition and mRNA destabilization in vitro and in vivo. Furthermore, our data suggest that at the apical tip of the testis *miR-9a* serves to disconnect GSCs from the hub by downregulating the expression of *N-cad*. GSCs adherent junctions are dynamic and constantly subjected to regulation. We propose

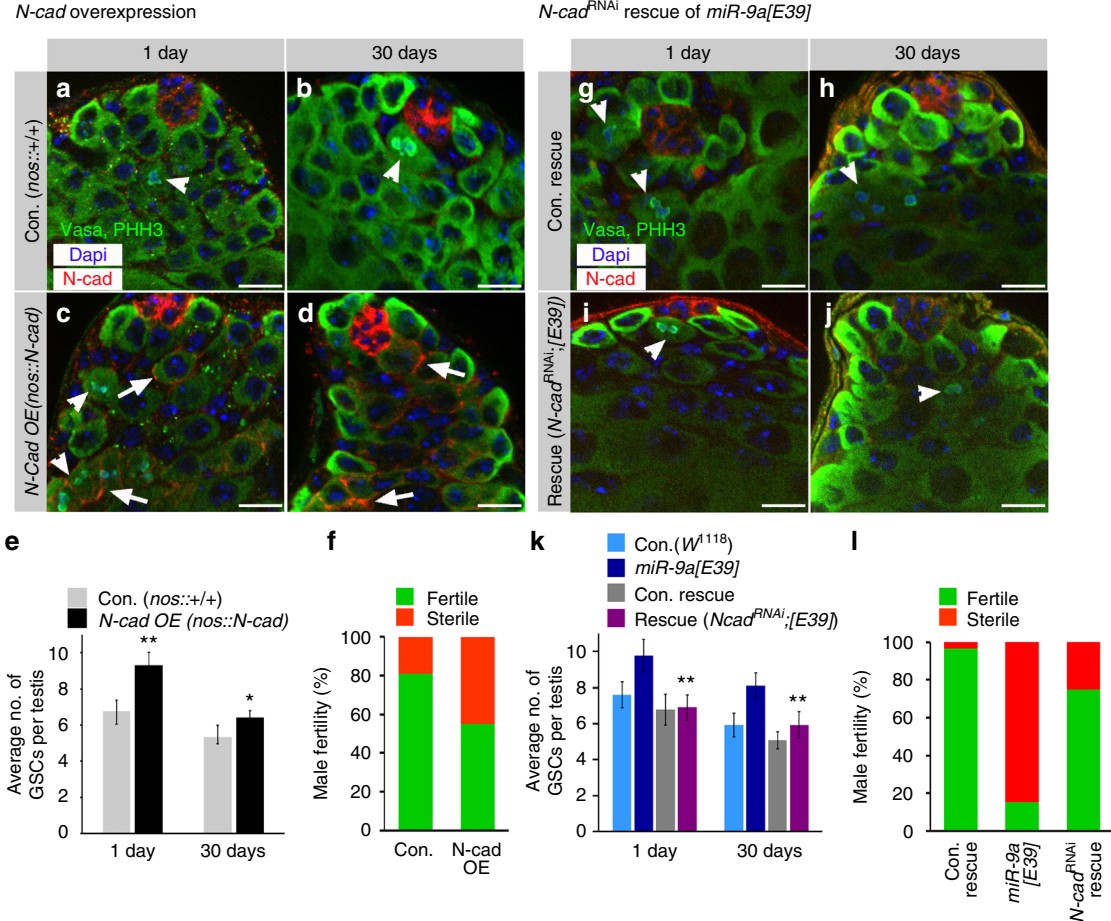

**Fig. 5** N-cad overexpression mimics *miR-9a[E39]* phenotype and N-cad[RNAi] rescues *miR-9a[E39]* mutants. **a–f** Overexpression of N-cad in GSCs and progenitor germ cells (N-cad OE: *nos-GAL4, UAS-N-cad*) increases GSCs number but reduces their division frequency compared to control (Con.: *nos-GAL4* outcrossed to *w[1118]*). **g–l** N-cad[RNAi] in GSCs and progenitor germ cells of *miR-9a[E39]* mutants rescue mutant phenotype (rescue: *nos-GAL4, UAS-N-cad[RNAi]; miR-9a[E39]*, Con. rescue: *nos-GAL4, UAS-N-cad[RNAi]; miR-9a[E39]*/TM6). **a–d, g–j** Representative images of testes immunostained for N-cad (*red*), Vasa and pHH3 (*green*) and DAPI (*blue; scale bars* 10 μm). *Arrowheads* denote mitotic germ cells and *arrows* denote N-cad ectopic expression (**c–d**). **e, k** Shown are average number of GSCs per testis along with 95% confidence interval (*error bars*). Statistical significance was determined as in Fig. 1d. **e** The total number of testes scored: Con. 1-day (*n* = 28) 30-days (*n* = 41); N-cad OE 1 -day (*n* = 35), 30-days (*n* = 41). *P* < 0.005 between N-cad OE and control in 1-day and *P* < 0.01 in 30-days. **k** The total number of testes scored: Con. (*w[1118]*) 1-day (*n* = 23), 30-days (*n* = 35); *miR-9a[E39]* 1-day (*n* = 27), 30-days (*n* = 40). Con. rescue 1-day (*n* = 27), 30 days (*n* = 29); *miR-9a* rescue 1-day (*n* = 30), 30-days (*n* = 31). *P* < 0.005 between N-cad[RNAi] *miR-9a[E39]* rescue and *miR-9a[E39]* in young and aged males. **f, l** Fertility assay of aged males. **f** Con. (*n* = 21) and N-cad OE (*n* = 21). Note decreased fertility of aged N-cad overexpression. **l** Con. rescue (*n* = 29), *miR-9a[E39]* (*n* = 21), and N-cad[RNAi] *miR-9a[E39]* rescue (*n* = 24).Note that N-cad[RNAi] rescues *miR-9a[E39]* sterility

that the differential expression of *miR-9a* in GSCs and progenitors, but not in hub cells, enables dynamic adherence of GSCs to the hub without affecting the hub cells themselves. We also suggest that the age-related increase in *miR-9a* levels correlates with the increase in dedifferentiation events of spermatogonia progenitors and possibly prevents accumulation of stem cells at the expense of spermatogenesis. In support of *miR-9a* as a negative regulator of adhesion are the findings that it represses two functionally orthologues of N-cad: in *Drosophila* the cadherin protein Flamingo (Fmi) and in mammals E-cad[22, 23], suggesting that *miR-9* plays a broad role in regulating cell–cell adhesion. Our transcriptome analysis revealed two additional adhesion proteins, *Sticks and Stones* and *lame duck* as potential targets of *miR-9a* (Table 1). Since N-cad is probably not the only mRNA target of *miR-9a*, it will be interesting to study whether these or other mRNA targets should be downregulated by *miR-9a* in order to regulate spermatogenesis during ageing.

In conclusion, we propose that the increase in *miR-9a* during ageing in stem and progenitor germ cells modulates degeneration in spermatogenesis and promotes detachment toward sperm maturation. The abnormal increase in stem cell maintenance in *miR-9a* mutant testis and the age-related loss of fertility illustrate the severe consequences of failure to restrain stem cell adherence to their niche.

## Methods

***Drosophila* stocks, ageing, and fertility**. Flies were raised at 25 °C on standard cornmeal molasses agar medium freshly prepared in our lab. Young flies were selected upon hatching and dissected in the first 3 days of their life. Young flies designated for ageing were placed in vials (20 males and 20 females per vial) that were replaced three times a week to prevent second generation from hatching and adult fly loss. Middle-aged flies were dissected at 15 days and aged flies at 30 days. Control and experiments were aged and tested at the same time. Fertility assays of aged males were performed by mating single males of each genotype (*n* ≥ 21) with three wild-type (*w[1118]*) virgin females. Males in vials that did not contain progeny were defined as sterile.

Fly strains used in this research were *w[1118]*, control GFP sensor (S.M. Cohen[18]), *miR-9a*-GFP sensor (E.C. Lai)[15], *upd-GAL4* (T. Xie), sco/cyo; *nos-Gal4:VP16/MKRS* (M. Van Doren[24]), *nos-GAL4/Cyo;Sb/Tm6b* (E. Arama), *Imp[CB04573]* (A. Spradling[25]), *UAS-DsRed-miR-9a* (E.C. Lai)[15], *UAS-DN-cadherin* on 2nd

(T. Uemura[26]), UAS-N-Cad[RNAi] (2[nd] Chr. VDRC #1092). miR-9a null mutant alleles (miR-9a[E39] (a backcrossed line) and miR-9a[J22]) were generated in Prof. Gao's lab by ends-out homologous recombination that replaced the 78 nt-long miR-9a precursor with the white gene[14]. To obtain identical genetic backgrounds that are critical for ageing experiments[27], miR-9a[E39] flies were first outcrossed in our lab to w[1118]. The siblings obtained (miR-9a[E39]/+) were then crossed again to obtain the miR-9a[E39]. Homozygosity was confirmed by the wing phenotype[15, 14] and PCR.

**Immunofluorescence**. Whole-mount testes from adult Drosophila were dissected in phosphate-buffered saline (PBS) and placed in Terasaki plates in 10 µl fix solution of 2% Paraformaldehyde (PFA) in PLP buffer (0.075 M lysine, 0.01 M sodium phosphate buffer, pH 7.4) for 1 h at room temperature, rinsed and washed twice in PBST (0.5% Triton X-100), followed by standard immunofluorescence staining. Primary antibodies used in this study were as follows: polyclonal rabbit anti-Vasa (Santa Cruz d-260, 1:200), rabbit anti-pHH3 (Merck Millipore 06-570, 1:200). Mouse anti-Fas3 (7G10, 1:10) and rat anti-DN-cad (DN-Ex #8, 1:5) were obtained from the Developmental Studies Hybridoma Bank at University of Iowa. Secondary antibodies were obtained from Jackson ImmunoResearch Laboratories Inc. Samples were mounted in Vectashield mounting medium with 4′,6-diamidino-2-phenylindole (DAPI; Vector Laboratories). Images were taken on a Zeiss Axio Observer microscope equipped with an Apotome system using the AxioVison software and were processed by Adobe Photoshop CS6.

**Stem cell counting and division frequency**. GSCs were counted as Vasa-positive cells that were in direct contact with the hub that was marked with anti-Fas3 or anti-DN-cad. Sample size of $n \geq 20$ testes was selected in preliminary studies to represent the average GSCs number and is indicted per each genotype and age. GSCs division frequency was calculated as the percentage of the total number of GSCs pHH3-positive cells from the overall GSCs number that were scored. For each study, results from more than 20 testes were pooled and averaged.

**miR-9a FISH**. Whole-mount testes were dissected in PBS diethyl pyrocarbonate (DEPC) and placed in Terasaki plates in 10 µl fix solution of 4% PFA in PBT buffer (PBS DEPC, 0.1% Tween-20) for 20 min at room temperature, rinsed and washed twice in PBTH (PBT, 50 µg/ml Heparin, and 250 µg/ml tRNA). Samples were rinsed once in Proteinase K buffer (5 mM Tris-HCl pH 7.4, 1 mM EDTA, 1 mM NaCl) and incubated for 3 min in 1 µl proteinase K diluted in Proteinase K buffer (20 mg/ml). Following washes in PBT samples were re-fixed and washed. Samples were then washed in Exiqon hybridization buffer and incubated for 1 h at 55 °C with 40 mM xtr-mir-9a-5p 5′ and 3′Dig labeled LNA probe (619314-360, Exiqon). Samples were then washed in serial dilutions of SSC buffer (5×, 1×, and 0.2× SSC) and blocked with TNB buffer (TSA Fluorescence Kit, PerkinElmer). Following overnight incubation with anti-DIG-POD antibody (Roche), signal was amplified with TSA Cyanine-3 and testes were mounted with DAPI.

**RNA extraction**. Testes of 100 flies (~200 testes) of each phenotype young and aged were dissected in PBS DEPC. Testes were collected and pooled in 100 µl TRIzol® Reagent and stored at −80 °C until future RNA extraction. To maximize RNA extraction, frozen samples were thawed at 37 °C and re-frozen in liquid nitrogen (−196 °C) five times, followed by five cycles of 30 s vortex and rest. Then 100 µl of 99% ethanol was added and the samples and total RNA was extracted using Direct-zol™ RNA miniprep kit (ZYMO Research) with DNase treatment, according to the manufacture instructions. The RNA was eluted in 50 µl of pre-heated DNase and RNase free water and kept in −80 °C for future use. RNA quality was measured by a bioanalyzer and samples were used for miRNAome, transcriptome, or qRT-PCR.

RNA extraction from S2R+ cells was obtained by Quick-RNA™ Miniprep kit (ZYMO Research), according to manufacture instructions. RNA was kept in RNase-DNase free water in −80 °C for until qRT-PCR analysis.

**Transcriptome analysis and statistics**. Illumina cDNA libraries were prepared from 1 µg total RNA extracted from testes of young and aged control w[1118] and miR-9a[E39] backcrossed mutants with TruSeq RNA V2/Illumina kit. Libraries were sequenced with Illumina HiSeq 2500. Raw reads were filtered for Illumina adapters and low-quality reads using Trimmomatic. Filtered reads were aligned to the Drosophila genome (dm6, FlyBase 6.05) using STAR (settings --alignIntronMax = 25,000 --genomeSAindexNbases = 9.15 --sjdbGTFfile). Gene expression levels were quantified using htseq-count, and differential expression was analyzed using edgeR. Differential expression data were filtered based on logFC ≥ 0.9, significance cutoff ($P$-value ≤ 0.05), and minimal reading levels (CPM ≥ 1). A group of genes that showed higher expression in miR-9a[E39] mutant vs. control in both young and aged testis was compared to a list of in silico predicted mir-9a targets (Targetscan Fly, http://www.targetscan.org/).

**miRNAome analysis and statistics**. Total purified RNA samples (0.5 µg at 100 ng/µl) of young, mid-aged, and aged w[1118] were used to determine the identity and levels of miRNAs with nanoString Technologies (nCounter fly miRNA expression kit). Raw data were normalized using the nCounter software and differential expression was analyzed using DESeq.

**Generation of DNA constructs**. The nucleotide numbers of the following sequences refer to their numbers in the 3′UTRs as shown on FlyBase. N-cad 3′UTR was amplified by PCR from genomic DNA with forward primer: N-cad3′ UTR_Xho1: CCGCTCGAGGCTGGTGGAGCGAGCAGTGATGAG, and reverse primer: N-cad 3′UTR_HInd3: CCCAAGCTTGCGCTCCTAAACTAAACTTT GGTATGCTCTC. PCR fragment was ligated into pGEM T Easy vector. To generate pGEM-N-cad 3′UTR-Mut nucleotides 180–182 ([180]CAA = TGG; Mut1) and 498–500 ([180]CAA = TGG; Mut2) were mutated by site-directed mutagenesis (Stratagene) with primers: nCad_3UTR_Mut1_For: CACACACACAAAGAAA GCGCAACTGGAGAAGCATCTCAATGCTG and nCad_3UTR_Mut1_Rev: CAGCATTGAGATGCTTCTCCAGTTGCGCTTTCTTTGTGTGTGTG, and nCad_3UTR_Mut2_For: GATTTGTATGATGTAGGGAGAGAGCATACTGG AGTTTAGTTTAGGAGCGC and nCad_3UTR_Mut2_Rev: GCGCTCCTAAAC-TAAACTCCAGTATGCTCTCTCCTACATCATACAAATC. pAc5-eGFP was a gift from D. Ideses (T. Juven-Gershon Lab) in which eGFP ORF was ligated via EcoRI and NotI into pAc5.1. N-cad-3′UTR[WT] or [Mut] was digested from pGEM with Xho1 and HInd3 and ligated into pAc5-eGFP to generate pAc-gfp-N-cad-3′ UTR[WT] or [Mut] that were used for transfection in S2R+ cells. The sequence of all DNA constructs described above was verified by DNA sequencing.

**S2R+ cell culture and transfection**. S2R+ adherent cells (T. Guven-Gershon) were cultured in Schneider's Drosophila medium supplemented with 10% heat-inactivated fetal bovine serum (Biological Industries Inc.) and 1% Penicillin Streptomycin. Each experiment was done at least in three biological repeats and each transfection was performed in triplicate. 600k cells were seeded in 24-well plates. pAc-GFP reporter plasmids (0.5 µg DNA per well) and dm-miR-9a mimic or negative control miR (10 nM; Applied Biosystems) were co-transfected according to the manufacturer's recommendations (LipoJet™). Post 48 h transfection cells were collected for flow cytometry, RNA/protein extraction.

**Flow cytometry**. Cells were washed twice and resuspended in 200 µl PBS. The samples were analyzed using BD FACSCanto II flow cytometer with DACSDiva software (BD Biosciences). Gates were set to exclude necrotic cells and cellular debris, and the fluorescence intensity of events within the gated regions was quantified. Data were collected from 10,000 events for each sample.

**Western blotting**. Cells were washed twice in PBS and total cell proteins were lyse with 1× Laemmli buffer. Proteins were separated on 12.5% SDS–PAGE gels, followed by western blotting according to standard procedures. Nitrocellulose membranes containing protein cell lysate incubated with primary antibodies at a dilution of 1:2000 rabbit anti-GFP (Cell Signaling; 2956) and 1:5000 mouse anti-Beta Actin (Sigma A5441). Proteins were visualized by a chemiluminescence detection kit for horseradish peroxidase (Biological Industries) and quantified using a CCD camera and Image-J software. The uncropped western blot can be found in Supplementary Fig. 4.

**qRT-PCR and statistics**. 1 µg RNA was treated with DNaseI (Promega) and reverse-transcribed with random hexamer mixture and the High-Capacity cDNA Reverse Transcription Kit (Thermo Fischer Scientific), according to the manufacture instructions. Quantitative real-time PCR was performed with a StepOne-Plus[TM] Real-time PCR System using TaqMan® Gene Expression Assay (Applied Biosystems). Relative gfp (assay ID: Mr04329676_mr) levels were compared to Ribosomal Protein L32 (RPL-32; assay ID: Dm02151827_g1). For miRNA analysis 10 ng RNA was used to prepare cDNA with TaqMan miRNA-specific reverse-transcription primers (Applied Biosystems) for miR-9a (Mir-9 assay ID: 000583) and ribosomal RNA 2S (V00236). Real-time PCR results were analyzed using StepOne software (Applied Biosystems) and significance was determined using Student's t-test. An average of three experiments (each performed in triplicate measurements) is shown (mean ± s.d.). P-values were generated after a two-tailed Student's t-test was used to compare $\Delta C_T$ between time points or genotypes across three independent biological replicates.

**Statistics**. For quantification of GSCs number and for densitometric analysis of pixel intensity, the mean ± 95% confidence interval and the number ($n$) of testes examined are shown. P-values were generated after a two-tailed Student's t-test or determined by one-way ANOVA, and post hoc analysis was performed with Tukey multi-comparison test if samples are normally distributed and have equal variances, **$P \leq 0.005$ and *$P \leq 0.01$.

**Data availability**. The transcriptome and miRNAome data have been deposited in Dryad Digital Repository (Epstein et al. 2017): DOI: http://dx.doi.org/10.5061/ dryad.30cn0.

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

## Acknowledgements

We are grateful to F.B. Gao, E.C. Lai, S.M. Cohen, S.F. Önel, T. Uemura, D. Ideses, T. Juven-Gershon, H.J. Bellen, O. Schuldiner, N. Issman, K. Yacobi-Sharon, and the Developmental Studies Hybridoma Bank for gracious gifts of *Drosophila* stocks, plasmids, and antibodies that were critical for this study. We thank N. Sher and the Bioinformatics unit of University of Haifa for transcriptome analysis. We also thank L. Barki-Harrington, D. Ideses, and H. Kaspi for advices and comments. This work was supported by the Israel Science Foundation (ISF) personal grant (1510/14), the Binational Science Foundation (BSF)- USA-Israel (2015398) and by Victoria and Marvin Sadowski Fund (Canada).

## Author contributions

Y.E., N.P., M.V., M.Z.F., R.B., L.P.K., and H.T. designed and carried out the experiments and interpreted the results. H.T. wrote the manuscript. N.P. and M.V. corrected the manuscript.

## Additional information

**Competing interests:** The authors declare no competing financial interests.

