## [Peer Review File · Nature Communications]

Reviewers' Comments:

Reviewer #1 (Remarks to the Author)

In this paper, Epstein et al. provide evidence that miR-9a regulates the proliferation of male germline stem cells by cell autonomously controlling the expression of N-cadherin. The authors first identify mir-9a as one of a handful of miRNAs whose testis expression increases as males age. They then show that germline stem cells in miR-9a mutant stem cells are elevated in number but divide less frequently, suggesting that they do not divide because they remain adhered to the hub. Then, based on differential gene expression analysis and some educated guessing, the authors focus in on N-cadherin as a likely target of miR-9a in germline stem cells. They show that N-cadherin seems to be more highly expressed in miR-9a mutant germline cells and that the N-cadherin 3'UTR contains sites that are responsive to miR-9a. Finally, the authors show that germline overexpression of N-cadherin phenocopies the miR-9a mutant phenotype. Overall, the data is well presented and the text is well written, although there are portions of the text that need additional clarification or re-writing. In addition, some of the experiments are missing key controls that are important for them to be convincing. Finally, the paper would be strengthened by experiments that directly address the central hypothesis regarding miR-9a function. These concerns are more fully elaborated in itemized comments below.

Major Comments

1. A better control for the sensor experiment (Fig 1e-g) would be the miR-9 sensor in a miR-9 mutant background. The control sensor and miR-9a sensor are from different labs and may have been differently constructed or inserted in different locations and these differences may explain the differences in expression levels. Furthermore, it is possible that the insertion of miR-9 sequences in the 3'UTR may destabilize the GFP transcript in a miR-9a-independent fashion.
2. The miR-9a phenotypes need to be rescued, either with a genomic fragment or a UAS-miR-9a transgene. Ideally, the authors would also show that additional miR-9a alleles also show the same germline stem cell phenotypes. Otherwise, these phenotypes could simply be due to background mutations in the control of miR-9 strains. Also, the authors mention that strains were outcrossed, but this should be more clearly described (how many backcrossings?).
3. The description of the gene expression analysis (e.g. lines 113-118) is confusing and needs to be described more clearly. First, is it really the case that 17,477 are differentially expressed? According to Flybase, the fly genome contains 17,728 genes, but it seems unlikely that essentially all genes are differentially expressed. Furthermore, it is also unclear whether this differential expression is between timepoints or between genotypes. Finally, the "additional filtration" (line 115) needs to be much more fully described to explain how the 17,477 were whittled down to 231.
4. The elevated N-cad expression in miR-9a mutants looks like it is also found between hub cells. How do the authors explain that if miR-9a is not expressed in the hub. To specifically label the elevated germline N-cad expression, could hub N-cad be knocked down in a miR-9a mutant?
5. In Fig 3c quantification, how was hub cell N-cad distinguished from germline N-cad?
6. Does overexpression of miR-9a in cell culture affect cell viability. This could explain the reduction in GFP expression shown in Fig 3.
7. The change in N-cad expression in 4c vs 4d is not entirely convincing, since the background looks higher in c than in d. The reduction could be clearer if miR-9a was driven with with both up

and nos-gal4's. How many representative are these images of miR-9a mutant testes?

8. Some kind of experiment showing that N-cad knockdown, either with RNAi or genetic mutants, rescues the miR-9a phenotypes is needed in order to convincingly argue that elevated N-cad is responsible for the miR-9a phenotype.

9. A more convincing experiment is needed to support the authors hypothesis that germline stem cells do not proliferate because they remain adhered to the hub. Could the authors generate MARCM-labeled miR-9 mutant germline stem cells. This should show that miR-9a mutant clones remain small and adherent to the hub, while control clones contain progenitor and differentiated cells distributed throughout the testis. Additionally, the authors could use this approach to knockdown N-cad in miR-9a clones to show rescue.

10. Is it possible that the miR-9a phenotype could be the result of elevated expression of other cell adhesion molecules like, for example, the previously identified E-cadherin Fmi? Is Fmi expressed in the male germline?

Minor Comments

Line 17/18: Should be rewritten, since it sounds like miR-9a is germline specific and not expressed in any other tissue.

Line 100/101: Numbers (n=) of samples analyzed should be included in the text here and also later when discussing the analysis of N-cadherin overexpression.

Line 107: Not all miRNA-mediated repression leads to mRNA decay.

Fig 2j/Fig 4j: The tables should be relabeled, since it is not clear what is listed on rows 2 and 3 (at least in the PDF version that I received).

Fig 2i,2k, 3c, 3f-3i, 4i, 4k: This is personal preference, but I'd recommend labeling significance of differences in the histograms using asterisks. They would be easier to see there than buried in the figure legends.

Reviewer #2 (Remarks to the Author)

The manuscript by Epstein et al. reports the function of miR-9a in germ cells during *Drosophila* aging. The authors found that miR-9a is upregulated in germline stem cells (GSCs) and spermatogonia progenitor cells during aging. miR-9a loss-of-function mutants have increased GSCs by staining, but decreased male fertility. The authors further characterized an increase of N-Cad upon miR-9a loss, and showed evidence supporting miR-9a directly targets N-Cad. Overexpression of N-Cad leads to similar phenotype as miR-9a loss-of-function mutant.

Although this story is potentially interesting to define a specific miRNA's function during aging in the germline, a number of deficiencies reduce enthusiasm. Major concerns include:

1. The genetic evidence is far from complete to support a miR-9a-N-Cad pathway in GSC aging. For example, the biological phenotype of miR-9a overexpression line was not described. The function of N-Cad as a downstream mediator of miR-9a activity is missing key rescue experiments. The authors should consider N-Cad knockdown in GSCs to rescue miR-9a E39 line. In addition, co-overexpression of miR-9a and N-Cad can be useful on top of the knockdown experiments.
2. It is evident from Fig 3b that there is increased N-Cad staining in hub cells upon miR-9a loss of function. It is difficult to judge whether N-Cad is increased in GSCs. If the authors believe N-Cad is increased in GSCs, more convincing evidence should be presented. Alternatively, the authors need to figure out why miR-9a loss-of-function mutants will result in increased N-Cad in hub cells, given

the claim by the authors that miR-9a is not expressed in hub cells.

3. it is critical that the phenotype of the E39 mutant is indeed due to loss of miR-9a function rather than other alterations at the locus. The original paper that published the E39 line also used another KO line J22. The authors can analyze J22 flies to determine their GSC phenotypes.

4. Although the authors attributed the decreased fertility of miR-9a loss-of-function mutant males to the defect in GSC proliferation, there lacks clear evidence to argue against alternative possibilities. For example, miR-9a is highly expressed in some other tissues beyond GSCs. One may argue that defects in those tissues lead to alterations of mating behavior and hence reduced fertility. I suggest the authors to do two things. First, if the authors' model is correct that the reduced fertility is due to lack of GSC proliferation, one would expect a reduced testis size and/or sperm count in miR-9a E39 aged males. Is this the case? Second, germline specific miR-9a overexpression rescue will be very useful to demonstrate a germ-tissue-specific contribution by miR-9a.

5. The increase of miR-9a in aged male germline is somewhat opposite to the expectation. The data would argue that miR-9a increase is helpful to maintain male fertility during aging. The gain-of-function experiments above will help to define the function of miR-9a increase. It could be interesting to discuss on this topic.

Minor:

1. Fig 2b: the genotype of the miR-9a E39 line is missing on the left.

2. Fig 2j, 3j: it is not fully clear what the second number row mean. I assume these are percentage of pH3 positive cells. I suggest the authors to explicitly label the rows in these two tables.

3. The p values for comparing the proliferating cells can use Fisher exact test rather than t-test. It is unclear to me how t-test can be used in this case.

Reviewer #3 (Remarks to the Author)

The manuscript by Epstein et al. explores the function of miR-9 in the *Drosophila* GSCs. The authors identify miR-9 from an expression screen performed during aging of *Drosophila* testis, where miR-9 accumulates with age. The authors use an elegant reporter system to unequivocally demonstrate that miR-9 is specifically expressed in GSCs and spermatogonia. The authors then show that loss of miR-9 function result in an increase in GSCs number, their reduced proliferation and ultimately reduced fertility that increases with severity upon age. The authors also identify N-cad as a target, validate it by reporter assays and in vivo both by loss and gain of miR-9 function. Finally, the authors demonstrate that over expression of N-cad in GSCs causes a similar phenotype to miR-9 mutants. In summary, N-cad is a plausible miR-9 target that could be at the source of the phenotype. Overall the manuscript is very well written and approachable to the broad readership of *Nature Communications*. The data presented are of high quality and the claims are supported by the data provided. From a stem cell, niche and ageing perspective the data are very interesting. I therefore endorse the manuscript for publication but have some concerns and questions that would need to be addressed prior.

Concerns:

1. Is the miR-9[E39] allele on the W111 genetic background? Can the authors confidently compare the impact on GSCs numbers and infertility?

2. Could the authors provide a brief description of the miR-9[E39] allele in the text? Is it a clean null allele and does the deletion affect neighboring loci?

3. I suggest removing the overexpression of miR-9 in the hub cells as the authors state that it was 'difficult and partial'. Given that miR-9 is not normally expressed in these cells it adds little to the manuscript and distracts from the clear phenotype in GSCs.

4. While the authors identify N-cad as a direct and likely important target of miR-9, it is unlikely to be the only one. The authors should present this important point in the discussion.

Response to Reviewers' comments on manuscript NCOMMS-16-18936-T

We would like to thank the Reviewers for their insightful comments and for their patience as we tested a number of reagents and protocols to address their concerns, often in aged genotypes. In this revised manuscript, we address all of the concerns raised by the reviewers, and provide a point-by-point rebuttal as detailed below.

Reviewer 1:

1. A better control for the sensor experiment (Fig 1e-g) would be the miR-9 sensor in a miR-9 mutant background. The control sensor and miR-9a sensor are from different labs and may have been differently constructed or inserted in different locations and these differences may explain the differences in expression levels. Furthermore, it is possible that the insertion of miR-9 sequences in the 3'UTR may destabilize the GFP transcript in a miR-9a-independent fashion.

This was an excellent suggestion, which we implemented and now appears in Supplementary Fig. 1a-b of the manuscript. Indeed expressing the *miR-9a* sensor in a *miR-9a* mutant background resulted in GFP expression in GSCs and spermatogonia cells, indicating that the sensor is specifically repressed by *miR-9a* in these cells. The *miR-9a* sensor result was further confirmed by *miR-9a* Fluorescence In Situ Hybridization (FISH) (mercury LNA detection probe *xtr-miR-9a-5p* 5' and 3' DIG labeled from Exiqon), which showed expression in GSCs and spermatogonia germ cells and not in the hub. Furthermore, there was no FISH signal in *miR-9a[E39]* mutants (Supplementary Fig. 1c-e). This information was added to the text in the last paragraph of page 3.

2. The miR-9a phenotypes need to be rescued, either with a genomic fragment or a UAS-miR-9a transgene.

We thank the Reviewer for suggesting this critical experiment. Overexpressing *UAS-miR-9a-DsRed* (Bejarano et al, 2010) in GSCs and spermatogonia cells of *miR-9a[E39]* mutants (*nosGal4,UAS-miR-9a-DsRed; miR-9a[E39]*) was sufficient to return the number of GSCs associated with the hub back to normal both in young and aged adults, increase the division rate of GSCs in aged males, and rescue the age-related sterility of *miR-9a[E39]* mutants. These results are depicted in Fig. 2 and described in the last paragraph of page 4 of the manuscript.

Ideally, the authors would also show that additional miR-9a alleles also show the same germline stem cell phenotypes. Otherwise, these phenotypes could simply be due to background mutations in the control of miR-9 strains.

Similar to *miR-9a[E39]* that was reported in the first version, we tested a second *miR-9a* null allele, *miR-9a[j22]* (Li et al., 2006), which we also found to display a high average number of GSCs in the niche of young (10.6 ± 0.7 , n=29) and aged males (8.4 ± 0.5 , n=17). Moreover, GSCs of aged *miR-9a[E39]* and *miR-9a[J22]* null mutants completely arrested division in aged flies. These results were added to the manuscript on page 4.

Also, the authors mention that strains were outcrossed, but this should be more clearly described (how many backcrossings?).

miR-9a[E39] was obtained from Prof. Gao's lab as a backcrossed line. These flies were first outcrossed in our lab to *w¹¹¹⁸*. The siblings obtained (*miR-9a[E39]/+*) were then crossed again to obtain the *miR-9a[E39]*. Homozygosity was confirmed by the wing phenotype (Li et al., 2006) and PCR. This explanation was added to the Methods, page 1.

3. The description of the gene expression analysis (e.g. lines 113-118) is confusing and needs to be described more clearly. First, is it really the case that 17,477 are differentially expressed? According to Flybase, the fly genome contains 17,728 genes, but it seems unlikely that essentially all genes are differentially expressed. Furthermore, it is also unclear whether this differential expression is between time points or between genotypes. Finally, the “additional filtration” (line 115) needs to be much more fully described to explain how the 17,477 were whittled down to 231.

We apologize for the confusing description provided in the original version and have now provided the following explanation on page 5 of the revised version:

"Reads were aligned to the *Drosophila* genome and gene expression levels were quantified using Htseq-count. This provided a list of 11,416 genes that are expressed in the testis (Supplementary Fig. 2). Differential gene analysis using the edgeR-classic method provided count per million (CPM) values and p-values. After filtration based on log Fold Change ($\log_{2}FC \geq 0.9$), significance cutoff ($p \text{ value} \leq 0.05$) and minimal CPM per each gene (≥ 1), we obtained a group of 450 genes that showed higher expression in young *mir-9a[E39]* mutant versus control, and 446 genes that were increased in old mutants versus control. Of these, 231 genes showed higher expression in *miR-9a[E39]* mutant versus control in both young and aged testis (Fig. 3a). A comparison of this list to the 194 *in-silico* predicted *miR-9a* targets (TargetsCan Fly) yielded six potential direct targets, one of which was *senseless*, a previously characterized target of *miR-9a*, confirming library reliability (Fig.3a and Table 1)". The software and filtration parameters are described in the Methods on page 3.

4. The elevated N-cad expression in miR-9a mutants looks like it is also found between hub cells. How do the authors explain that if miR-9a is not expressed in the hub. To specifically label the elevated germline N-cad expression, could hub N-cad be knocked down in a miR-9a mutant?

We thank the Reviewer for this comment, as it clarified that we failed to properly explain the niche architecture. The hub is a spherical 3-dimensional (3D) structure of approximately 12 cells, the great majority of which (~9-10) are associated on several planes with **all** the surrounding GSCs (~8). Furthermore, the size of the hub cells is considerably smaller than that of the GSCs. However, depiction of this data in a 2D image may appear as though many more hub cells are in contact only among themselves than in reality. Therefore, what may appear as an increase in N-cad among hub cells following *miR-9a* loss is in fact an increase that occurs between hub and GSCs. To depict this point we added a series of Z-stack images of the image presented in Fig. 3b, which shows that GSCs pile around the hub sphere on several planes (Supplementary Fig. 3a-a"). We also presented a 3D projection of 10 Z-stacks showing the spherical structure of the niche (Supplementary Fig. 3b). To further strengthen our hypothesis that N-cad adherent junctions are mostly found in GSCs-hub boundaries, we reduced its expression in the GSCs of the *miR-9a[E39]* mutants (*nosGal4,UAS-N-cad^{RNAi};miR-9a[E39]*). As shown in Supplementary Fig. 3c-d, this resulted in overall reduction of N-cad staining. This experiment is described on the first paragraph of page 6. Additional phenotypes of this experiment are entailed in Section 8 of this rebuttal. An explanation about hub architecture was added to the Introduction on page 2.

5. In Fig 3c quantification, how was hub cell N-cad distinguished from germline N-cad?

Quantification of N-cad expression was done by defining the entire hub domain. As explained in the previous item, the great majority of the hub cells are in contact with GSCs (*i.e.* maintain hub-GSC and hub-hub boundary) whereas only a 2-3 cells have only hub-hub contacts. Therefore, most of the observed increase in N-cad levels is related to the loss of *miR-9a* in the germline.

6. Does overexpression of miR-9a in cell culture affect cell viability. This could explain the reduction in GFP expression shown in Fig 3.

The Reviewer is correct in that overexpression may sometimes affect cell viability. However, this is not the case with *miR-9a* overexpression in S2R+ cell culture as no exceptional amounts of dead cells were observed compared to the controls in either the flow experiments or under the microscope. Moreover, as depicted in Fig. 3e, *miR-9a* is also overexpressed together with N-cad 3'UTR Mut (lane 4), which did not affect the expression of GFP. The transfection efficiency of *miR-9a* co-expression with GFP-*N-cad-3'UTR^{WT}* or GFP-*N-cad-3'UTR^{Mut}* reporters was also measured by qRT-PCR for mature *miR-9a*, and confirmed similar expression levels. A comment regarding this issue was added to the manuscript on page 6.

7. The change in N-cad expression in 4c vs 4d is not entirely convincing, since the background looks higher in c than in d. The reduction could be clearer if miR-9a was driven with both *upd* and *nos-gal4*'s. How many representatives are these images of miR-9a mutant testes?

Overexpressing *UAS-miR-9a-DsRed* (Bejarano et al, 2010) in the hub (*updGal4,UAS-miR-9a-DsRed*) was partial in the sense that we could clearly detect a DsRed signal in only a few samples (8/21). However, in those samples where DsRed expression was detected N-cad expression decreased dramatically. We now provide two representative images (Fig. 4d-e) relative to control (Fig. 4c), all taken at the same exposure time.

Additionally, we performed the experiment requested by the Reviewer where *UAS-miR-9a-DsRed* was simultaneously overexpressed in both the hub and GSCs (*updGal4;nosGal4,UAS-miR-9a-DsRed*). This caused a dramatic reduction in N-cad expression (Fig. 4g). However, here too clear detection of the DsRed signal in both hub and GSCs was only apparent in part of the samples (14/39, Fig. 4g) while the rest of the samples expressed *miR-9a-DsRed* only in the germline (25/39, Fig. 4f). These results are described in page 6 of the manuscript and in Fig. 4.

8. Some kind of experiment showing that N-cad knockdown, either with RNAi or genetic mutants, rescues the miR-9a phenotypes is needed in order to convincingly argue that elevated N-cad is responsible for the miR-9a phenotype.

Again, we thank the Reviewer for suggesting this critical experiment. Reducing N-cad levels with *UAS-N-cad^{RNAi}* in GSCs and spermatogonia cells of *miR-9a[E39]* mutants (*nosGal4,UAS-N-cad^{RNAi};miR-9a[E39]*) was sufficient to restore a normal average number of GSCs, division rate and fertility. These results are described in page 7 of the manuscript and in Fig 5.

9. A more convincing experiment is needed to support the authors hypothesis that germline stem cells do not proliferate because they remain adhered to the hub. Could the authors generate MARCM-labeled miR-9 mutant germline stem cells. This should show that miR-9a mutant clones remain small and adherent to the hub, while control clones contain progenitor and differentiated cells distributed throughout the testis. Additionally, the authors could use this approach to knockdown N-cad in miR-9a clones to show rescue.

The Reviewer is correct in his/her comment that MARCM *miR-9a* mutant clones could further support our hypothesis and we have attempted to perform the suggested experiment. However, we encountered significant technical issues in obtaining the triple recombinant flies that we designed for the MARCM crosses (*miR-9a[E39], FRT2A, nanos-GAL4*). However, in this revised manuscript we used the other critical experiments suggested by the Reviewers to strengthen our hypothesis by showing that GSCs of *miR-9a* mutants arrest division in aged

flies (PHH3 staining and fertility assay) and that these phenotypes are completely rescued either by *N-cad*^{RNAi} or *miR-9a* overexpression in GSCs.

10. Is it possible that the *miR-9a* phenotype could be the result of elevated expression of other cell adhesion molecules like, for example, the previously identified E-cadherin *Fmi*? Is *Fmi* expressed in the male germline?

Our transcriptome analysis of testes from wild-type and *miR-9a*[E39] mutants shows that *Fmi* levels are low and are not further increased in the mutant flies (See Reviewer table 1 below). Moreover, staining the testis with anti-*Fmi* Ab (Hybridoma Bank) did not reveal any signal, although staining embryos in the same sample showed *Fmi* expression in the CNS (See Reviewer Fig. 1 below). These results suggest that in the testis, the *miR-9a* phenotype is not due to elevated expression of *Fmi*.

Gene	1d-Young				30d-Aged				Seeds
	logFC	Significance (p value)	Average CPM per gene		logFC	Significance (p value)	Average CPM per gene		
			WT	Mir-9a mutant			WT	Mir-9a mutant	
N-cadherin	1.6	1.3E ⁻²⁶	20.7	63.4	0.9	4.9E ⁻⁰⁸	27.0	49.3	2
Fmi (starry night)	-0.2	0.53	5.5	4.9	-1.2	0.0002	2.9	1.3	1

Reviewer Table 1: Transcriptome analysis of *N-cadherin* and *Fmi* in testis of young and aged wild-type and *miR-9a*[E39] mutants. *Fmi* levels are low and not increase in *miR-9a* mutants.

Reviewer Fig. 1. Apical tip of the testis (left) and embryo stage 14 (right, arrow marks anterior) stained with *Fmi* (red), *Vasa* (green) and DAPI (blue). *Fmi* is expressed in the embryo CNS but not in the testis. Scale-bars 20µm.

Minor Comments

Line 17/18: Should be rewritten, since it sounds like *miR-9a* is germline specific and not expressed in any other tissue.

The sentence was revised to show that the described phenomenon pertains to the *Drosophila* testis: "Here we report that in the *Drosophila* testis, the conserved *miR-9a* is expressed in germline stem and progenitor cells and its levels are significantly elevated during ageing."

Line 100/101: Numbers (n=) of samples analyzed should be included in the text here and also later when discussing the analysis of *N-cadherin* overexpression.

The number of GSCs that were scored and the percentage of PHH3 positive GSCs were added to the text on pages 4 and 7.

Line 107: Not all miRNA-mediated repression leads to mRNA decay.

The sentence was changed to: "miRNAs repress mRNA translation, which is often followed by the mRNA deadenylation and decay."

Fig 2j/Fig 4j: The tables should be relabeled, since it is not clear what is listed on rows 2 and 3 (at least in the PDF version that I received).

We apologize for the quality of the PDF in general and particularly for the data that was presented in the Tables in Fig. 2 and 4. We realize that presentation of the data in this manner was cumbersome. We therefore removed these Tables from the revised version and include the data regarding mitotic GSCs and the overall GSCs that were scored in the text on pages 4 and 7.

Fig 2i,2k, 3c, 3f-3i, 4i, 4k: This is personal preference, but I'd recommend labeling significance of differences in the histograms using asterisks. They would be easier to see there than buried in the figure legends.

The Reviewer is correct and asterisks were added to the histograms.

Reviewer #2 (Remarks to the Author):

1. The genetic evidence is far from complete to support a miR-9a-N-cad pathway in GSC aging. For example, the biological phenotype of miR-9a overexpression line was not described. The function of N-cad as a downstream mediator of miR-9a activity is missing key rescue experiments. The authors should consider N-cad knockdown in GSCs to rescue miR-9a E39 line. In addition, co-overexpression of miR-9a and N-cad can be useful on top of the knockdown experiments.

We thank the Reviewer for this insightful comment. While overexpression of *miR-9a* in the germline of WT flies (*nos-GAL4;UAS-miR-9a-DsRed*) reduces N-cad expression (Fig. 4), it does not show a significant effect on the number of GSCs in the aged fly. On the other hand, overexpression of *miR-9a* on a mutant background (*nosGal4,UAS-miR-9a-DsRed;miR-9a[E39]*) restores the average stem cell number back to their original numbers in the aged fly (Fig. 2). The reason for which the *miR-9a* overexpression in the WT does not show any effect probably stems from the fact that aged animals already express very high amounts of *miR-9a* (up to ~1% of the entire miRNA in the testis; a comment was added to the results on page 3), this may generate a 'ceiling effect' where additional overexpression cannot enhance the effect further. Having said that, we performed the key experiments suggested by the Reviewer, which considerably strengthen the genetic evidence for *miR-9a* - N-cad axis in aging. We now show that reducing *N-cad* levels in GSCs and spermatogonia cells of *miR-9a[E39]* mutants (*nosGal4,UAS-N-cad^{RNAi};miR-9a[E39]*) was sufficient to: **a)** return the number of GSCs associated with the hub back to normal numbers in both young and aged adults **b)** increase the division rate of GSCs in aged males, and **c)** rescue the age-related sterility of *miR-9a[E39]* mutants (See Fig. 5 and page 7 in the manuscript).

2. It is evident from Fig 3b that there is increased N-cad staining in hub cells upon miR-9a loss of function. It is difficult to judge whether N-cad is increased in GSCs. If the authors believe N-cad is increased in GSCs, more convincing evidence should be presented. Alternatively, the authors need to figure out why miR-9a loss-of-function mutants will

result in increased N-cad in hub cells given the claim by the authors that miR-9a is not expressed in hub cells.

This is an important point that we now clarify. First, by adding a FISH experiment for *miR-9a*, we strengthen the data obtained from the *miR-9a*-GFP sensor (Fig. S1) that indeed *miR-9a* is expressed in GSCs and spermatogonia but not in hub cells. Second, the hub is a spherical 3-dimensional (3D) structure of approximately 12 cells, the great majority of which (~9-10) are associated on several planes with **all** the surrounding GSCs (~8). Furthermore, the size of the hub cells is considerably smaller than that of the GSCs. However, depiction of this data in a 2D image may appear as though many more hub cells are in contact only among themselves than in reality. Therefore, what may appear as an increase in N-cad among hub cells following *miR-9a* loss is in fact an increase that occurs between hub and GSCs. To depict this point we added a series of Z-stack images of the image presented in Fig. 3b, which shows that GSCs pile around the hub sphere in several plans (Supplementary Fig. 3a-a"). We also presented a 3D projection of 10 Z-stacks showing the spherical structure of the niche (Supplementary Fig. 3b). To further strengthen our hypothesis that the major role of N-cad is to adhere stem and niche, we stained the described above *N-cad^{RNAi}* in the *miR-9a[E39]* mutants (*nosGal4,UAS-N-cad^{RNAi};miR-9a[E39]*) with anti-N-cad antibodies. As shown in Supplementary Fig. 3c-d, this resulted in an overall reduction of N-cad staining. This experiment is described on the first paragraph of page 6 of the manuscript. An explanation about hub architecture was added to the Introduction on page 2.

3. It is critical that the phenotype of the E39 mutant is indeed due to loss of miR-9a function rather than other alterations at the locus. The original paper that published the E39 line also used another KO line J22. The authors can analyze J22 flies to determine their GSC phenotypes.

We thank the Reviewer for suggesting these experiments to further strengthen our data. To verify that the phenotypes are due to *miR-9a* loss we did the following:

- 1) *miR-9a[E39]* was obtained from Prof. Gao's lab as a backcrossed line. These flies were then outcrossed in our lab first to *w¹¹¹⁸*. The siblings obtained (*miR-9a[E39]/+*) were then crossed again to obtain the *miR-9a[E39]*. Homozygosity was confirmed by the wing phenotype^{2 3} and PCR. This explanation was added to Methods, page 1.
- 2) Similar to *miR-9a[E39]*, we tested the second *miR-9a* null allele, *miR-9a[J22]* (Li et al., 2006), which was also found to maintain a high average number of GSCs in the niche of young (10.6 ± 0.7 , n=29) and aged males (8.4 ± 0.5 , n=17). Moreover, GSCs of aged *miR-9a[E39]* and *miR-9a[J22]* null mutants completely arrested division in aged flies. These results were added to the manuscript on page 4.

4. Although the authors attributed the decreased fertility of miR-9a loss-of-function mutant males to the defect in GSC proliferation, there lacks clear evidence to argue against alternative possibilities. For example, mir-9a is highly expressed in some other tissues beyond GSCs. One may argue that defects in those tissues lead to alterations of mating behavior and hence reduced fertility. I suggest the authors to do two things. First, if the authors' model is correct that the reduced fertility is due to lack of GSC proliferation, one would expect a reduced testis size and/or sperm count in miR-9a E39 aged males. Is this the case? Second, germline specific miR-9a overexpression rescue will be very useful to demonstrate a germ-tissue-specific contribution by miR-9a.

The Reviewer is correct in his/her comment that lack of fertility in aged *miR-9a* mutants can result from additional factors besides a GSC proliferation defect. Since the sperm cells in the *Drosophila* testis are long, convoluted and attached to one another, we were unable to obtain

a reliable measurement. We therefore took the Reviewers second advice and selectively overexpressed *miR-9a* in the GSCs and spermatogonia cells of the *miR-9a[E39]* mutants. The results, which were added to Fig. 2, clearly indicate that this was sufficient to completely rescue the age-related sterility.

5. The increase of *miR-9a* in aged male germline is somewhat opposite to the expectation. The data would argue that *miR-9a* increase is helpful to maintain male fertility during aging. The gain-of-function experiments above will help to define the function of *miR-9a* increase. It could be interesting to discuss on this topic.

As noted in the section above, overexpression of *miR-9a* in the mutants not only rescued fertility but also reduced the number of GSCs back to normal and regained their ability to differentiate. We too were initially surprised that an increase in GSCs was not translated into increased spermatogenesis. However, we show that normal spermatogenesis depends on the ability of the stem cells to adequately detach from the niche, a process that is regulated by the *miR-9a-N-cad* pathway. This point was further elaborated on in the Discussion on page 8.

Minor:

1. Fig 2b: the genotype of the *miR-9a E39* line is missing on the left.

2. Fig 2j, 3j: it is not fully clear what the second number row mean. I assume these are percentage of pHH3 positive cells. I suggest the authors to explicitly label the rows in these two tables.

We apologize for the quality of the PDF in general and particularly for the data that was presented in the Tables in Fig. 2 and 4. We realize that presentation of the data in this manner was cumbersome. We therefore removed these Tables from the revised version and include the data regarding mitotic GSCs and n number of GSCs that were scored in the text on pages 4 and 7.

3. The p values for comparing the proliferating cells can use Fisher exact test rather than t-test. It is unclear to me how t-test can be used in this case.

The *t*-test and p value data that was described in the Methods section of manuscript refers only to the data presented in the graphs, where normal distribution was confirmed prior to the statistical test. As mentioned in the section above, we replaced the table of proliferating GSCs with text. Since we obtain our images at a specific time point, we essentially gain a snapshot of dividing GSCs, an event that occurs on average once every 24h (~6% of the cells). This is further reduced during aging (down to ~3% of the cells). These results are in agreement with the findings of Cheng et al., Nature 2008 (Ref 8 of the manuscript). However, in aged *miR-9a* mutants this low number is reduced to zero dividing cells since there was no detection of pHH3 in 233 and 178 scored GSCs of *miR-9a[E39]* and *miR-9a[J22]*, respectively. Together this suggests that the data is more of qualitative than quantitative nature.

Reviewer #3 (Remarks to the Author):

Concerns:

1. Is the *miR-9[E39]* allele on the *w¹¹¹⁸* genetic background? Can the authors confidently compare the impact on GSCs numbers and infertility *miR-9[E39]*?

The *miR-9[E39]* backcrossed line was obtained from Prof. Gao's lab and outcrossed in our lab to *w¹¹¹⁸* in order to obtain the same genetic background. The homozygosity of *miR-9[E39]* was confirmed by the wing phenotype (Li et al., 2006 and Bejarano et al, 2010) and PCR. This explanation was added to the Methods page 1. Therefore, the comparison between the

two lines in terms of GSC number and fertility is valid. These results are further strengthened by an additional observation of rescue of GSC and fertility following overexpression of *miR-9a* in stem and progenitor cells of *miR-9a[E39]* mutants. These experiments were added to the manuscript on the last paragraph of page 4 and Fig. 2.

2. *Could the authors provide a brief description of the miR-9[E39] allele in the text? Is it a clean null allele and does the deletion affect neighboring loci?*

The *miR-9[E39]* loss of function mutant was generated in Prof. Gao's lab by ends-out homologous recombination that replaced the 78-nt long *miR-9a* precursor with the *white* gene (Li et al., 2006). Thus the *miR-9[E39]* null mutant is indeed a clean null allele, as confirmed by Li et al., 2006 (Ref 13 in the manuscript). Furthermore, as indicated in the previous section, outcrossing this allele to *w¹¹¹⁸* further cleaned the genetic background. We added a short description of the mutant and outcross to the Methods on page 1.

3. *I suggest removing the overexpression of miR-9 in the hub cells as the authors state that it was 'difficult and partial'. Given that miR-9 is not normally expressed in these cells it adds little to the manuscript and distracts from the clear phenotype in GSCs.*

We thank the Reviewer for this comment, as it clarified that we failed to properly explain what we meant by "difficult and partial". N-cad is a protein that forms homophilic interactions. Therefore, to understand its function in GSC-hub interactions, we felt it was important to study the effect of its knockdown through *miR-9a* overexpression (*UAS-miR-9a-DsRed*, Bejarano et al, 2010) in both directions. While overexpression of *miR-9a* in the GSCs yielded DsRed signal in all samples, its overexpression in the hub was difficult on the flies as only very few showed any DsRed signal. However, in samples that did show such a signal, the expression of N-cad was dramatically reduced. If the Reviewer still feels that this explanation is distracting, we are happy to move this data set to the Supplementary section or completely remove it from the manuscript.

4. *While the authors identify N-cad as a direct and likely important target of miR-9, it is unlikely to be the only one. The authors should present this important point in the discussion.*

Again, we thank the Reviewer for highlighting this critical point. We now clarify the fact that *N-cad* is only one *miR-9a* mRNA target among several that should be downregulated in stem cells in order to maintain normal spermatogenesis. A comment regarding this matter was added to the end of the Discussion on page 8.

Reviewers' Comments:

Reviewer #1:

Remarks to the Author:

This version of the manuscript is significantly improved, especially with the addition of the N-cad RNAi rescue data at the end of the results. However, I still have two issues that I think should be addressed.

Major comments

1. I am not convinced that the title of the paper accurately conveys the results in the paper. The data in the paper clearly shows that miR-9 has a GSC defect but it is not clear whether the miR-9 phenotype worsens with age. Even young miR-9 mutants have a GSC defect so miR-9 does not solely have an ageing function. The authors suggest that GSC mitotic activity in miR-9 mutants worsens with age, but they don't report PH3 number in young flies so it is hard to know. Without additional data, the authors should consider changing the title

2. The data in Fig 3B looks like loss of miR-9 leads to N-Cad increase in the hub. The authors suggest that it may actually be an increase in the GSCs (line 165). This issue could be directly addressed if the authors test the cell autonomy of miR-9 either by using the miR-9 sponge to show that miR-9 knockdown in the germline but not the hub is associated with elevated N-Cad or that GSC clones of miR-9 contain more N-Cad. The additional experiments in Fig S3C,D and 5K are good, but they don't rule out the possibility that knockdown of N-Cad in the germline could lead to cell non-autonomous reduction of N-cad in the hub.

Minor comments

line 71: remove the word three unless you explain how the candidates were ranked relative to one another

line 72: change "abounded" to "abundant"

line 83: insert "the" between "GFP of" and "control"

line 163: change "plans" to "planes"

Reviewer #2:

Remarks to the Author:

The revised manuscript has addressed most of my concerns. Particularly, the genetic rescue experiments with miR-9a and N-Cad have significantly strengthened the conclusion.

One weakness, however, is not fully addressed---the authors concluded that the regulation of N-cad by miR-9a is a direct regulation occurring in GSCs. However, this still does not have strong support from cellular staining, even with the new figures in supplement Fig 3. It is difficult to exclude a possibility that miR-9a's major function in GSCs is to regulate N-cad in hub cells, possibly through other targets. The authors should at least explicitly discuss this possibility in the discussion.

Another minor issue is for the J22 line of miR-9a loss-of-function mutant. The authors only gave a brief description in text, without mentioning control levels. The authors can show a figure, possibly in supplement, on these data, with proper controls.

Reviewer #3:

Remarks to the Author:

My concerns have been addressed. I endorse the the manuscript for publication.

Response to Reviewers' comments on manuscript NCOMMS-16-18936A

We would like to thank the Reviewers for reviewing the revised manuscript. We further address all of the concerns raised and provide a point-by-point rebuttal as detailed below.

Reviewer #1 (Remarks to the Author):

This version of the manuscript is significantly improved, especially with the addition of the N-cad RNAi rescue data at the end of the results. However, I still have two issues that I think should be addressed.

Major comments

1. I am not convinced that the title of the paper accurately conveys the results in the paper. The data in the paper clearly shows that miR-9 has a GSC defect but it is not clear whether the miR-9 phenotype worsens with age. Even young miR-9 mutants have a GSC defect so miR-9 does not solely have an ageing function. The authors suggest that GSC mitotic activity in mir-9 mutants worsens with age, but they don't report PH3 number in young flies so it is hard to know. Without additional data, the authors should consider changing the title

The Reviewer is correct in that young *miR-9a* mutants already have a GSC defect. However, this defect is strengthened during ageing and eventually arrests spermatogenesis (Fig. 2n). Therefore, we changed the title to: "*miR-9a* modulates maintenance and ageing of *Drosophila* germline stem cells by limiting the expression of N-cadherin".

2. The data in Fig 3B looks like loss of miR-9 leads to N-Cad increase in the hub. The authors suggest that it may actually be an increase in the GSCs (line 165). This issue could be directly addressed if the authors test the cell autonomy of miR-9 either by using the miR-9 sponge to show that miR-9 knockdown in the germline but not the hub is associated with elevated N-Cad or that GSC clones of miR-9 contain more N-Cad. The additional experiments in Fig S3C,D and 5K are good, but they don't rule out the possibility that knockdown of N-Cad in the germline could lead to cell non-autonomous reduction of N-cad in the hub.

We agree with the Reviewer that despite our data, we cannot completely rule out the possibility that reduction of *N-cad* in the germline, may potentially lead to non-autonomous reduction of N-cad in the hub. This is despite an additional support of our findings (Supplementary Fig. 3 C-D and Fig. 5K) by a paper that shows that GSCs project microtubule-based nanotubes, which extend and protrude into the hub (described on page 2, Inaba *et al.*, Nature 2015; Ref. 6). Nonetheless, following the Reviewer's suggestion to perform the *miR-9a* sponge experiment we have contacted Prof. David Van Vactor at Harvard Medical School, who created the complete miRNA sponge transgenic collection in his lab (Fulga *et al.*, Nat Communications 2015). In a personal communication Prof. Van Vactor informed us that the *miR-9a* sponge is not working for unknown technical reasons, which prevented him from

depositing this specific line to the Bloomington Drosophila Stock Center. Although this line can be reconstructed, this is a lengthy process of approximately 12 months that ultimately will not substantially change the main findings of the paper. Having said that, we accept the comment and have now included an extended discussion of the non-autonomous possibility on page 8.

Minor comments

line 71: remove the word three unless you explain how the candidates were ranked relative to one another

line 72: change “abounded” to “abundant”

line 83: insert “the” between “GFP of” and “control”

line 163: change “plans” to “planes”

All requested changes were incorporated into the text.

Reviewer #2 (Remarks to the Author):

The revised manuscript has addressed most of my concerns. Particularly, the genetic rescue experiments with miR-9a and N-Cad have significantly strengthened the conclusion.

One weakness, however, is not fully addressed---the authors concluded that the regulation of N-cad by miR-9a is a direct regulation occurring in GSCs. However, this still does not have strong support from cellular staining, even with the new figures in supplement Fig 3. It is difficult to exclude a possibility that miR-9a’s major function in GSCs is to regulate N-cad in hub cells, possibly through other targets. The authors should at least explicitly discuss this possibility in the discussion.

We agree with the Reviewer and now explicitly discuss the possibility of regulating N cad in the hub via non-autonomous targets in the Discussion on page 8.

Another minor issue is for the J22 line of miR-9a loss-of-function mutant. The authors only gave a brief description in text, without mentioning control levels. The authors can show a figure, possibly in supplement, on these data, with proper controls.

At the Reviewer's request, the data regarding the *miR-9a*[J22] line and control were added to the Supplemental information in Supplementary Fig. 1f.

Reviewer #3 (Remarks to the Author):

My concerns have been addressed. I endorse the manuscript for publication.